# Genetic Dominant Variants in *STUB1,* Segregating in Families with SCA48, Display In Vitro Functional Impairments Indistinctive from Recessive Variants Associated with SCAR16

**DOI:** 10.3390/ijms22115870

**Published:** 2021-05-30

**Authors:** Yasaman Pakdaman, Siren Berland, Helene J. Bustad, Sigrid Erdal, Bryony A. Thompson, Paul A. James, Kjersti N. Power, Ståle Ellingsen, Martin Krooni, Line I. Berge, Adrienne Sexton, Laurence A. Bindoff, Per M. Knappskog, Stefan Johansson, Ingvild Aukrust

**Affiliations:** 1Department of Medical Genetics, Haukeland University Hospital, 5021 Bergen, Norway; yasaman.pakdaman@uib.no (Y.P.); siren.berland@helse-bergen.no (S.B.); sigrid.erdal@helse-bergen.no (S.E.); per.morten.knappskog@helse-bergen.no (P.M.K.); Stefan.Johansson@uib.no (S.J.); 2Department of Clinical Science, University of Bergen, 5021 Bergen, Norway; 3Department of Biomedicine, University of Bergen, 5021 Bergen, Norway; helene.bustad.johannessen@gmail.com; 4Department of Pathology, Royal Melbourne Hospital, Parkville, VIC 3050, Australia; Bryony.Thompson@mh.org.au; 5Department of Clinical Pathology, Melbourne Medical School, University of Melbourne, Parkville, VIC 3010, Australia; 6Genomic Medicine, Royal Melbourne Hospital, Parkville, VIC 3050, Australia; Paul.James@petermac.org (P.A.J.); adrienne.sexton@mh.org.au (A.S.); 7Department of Medicine, University of Melbourne, Parkville, VIC 3051, Australia; 8Department of Neurology, Haukeland University Hospital, 5021 Bergen, Norway; kjersti.nesheim.power@helse-bergen.no; 9Department of Biological Sciences, University of Bergen, 5006 Bergen, Norway; Stale.Ellingsen@uib.no; 10NKS Olaviken Gerontopsychiatric Hospital, 5306 Askøy, Norway; martin.krooni@gotland.se (M.K.); Line.Berge@uib.no (L.I.B.); 11Department of Health and Medical Care, Clinic of Psychiatry, Visby Hospital, 621 55 Visby, Sweden; 12Centre for Elderly and Nursing Home Medicine, Department of Global Public Health and Primary Care, University of Bergen, 5009 Bergen, Norway; 13Neuro-SysMed, Center of Excellence for Clinical Research in Neurological Diseases, Department of Neurology, Haukeland University Hospital, 5021 Bergen, Norway; laurence.albert.bindoff@helse-bergen.no; 14Department of Clinical Medicine, University of Bergen, 5021 Bergen, Norway

**Keywords:** spinocerebellar ataxia, SCA48, SCAR16, CHIP, STUB1, E3 ubiquitin ligase

## Abstract

Variants in *STUB1* cause both autosomal recessive (SCAR16) and dominant (SCA48) spinocerebellar ataxia. Reports from 18 *STUB1* variants causing SCA48 show that the clinical picture includes later-onset ataxia with a cerebellar cognitive affective syndrome and varying clinical overlap with SCAR16. However, little is known about the molecular properties of dominant *STUB1* variants. Here, we describe three SCA48 families with novel, dominantly inherited *STUB1* variants (p.Arg51_Ile53delinsProAla, p.Lys143_Trp147del, and p.Gly249Val). All the patients developed symptoms from 30 years of age or later, all had cerebellar atrophy, and 4 had cognitive/psychiatric phenotypes. Investigation of the structural and functional consequences of the recombinant C-terminus of HSC70-interacting protein (CHIP) variants was performed in vitro using ubiquitin ligase activity assay, circular dichroism assay and native polyacrylamide gel electrophoresis. These studies revealed that dominantly and recessively inherited *STUB1* variants showed similar biochemical defects, including impaired ubiquitin ligase activity and altered oligomerization properties of the CHIP. Our findings expand the molecular understanding of SCA48 but also mean that assumptions concerning unaffected carriers of recessive *STUB1* variants in SCAR16 families must be re-evaluated. More investigations are needed to verify the disease status of SCAR16 heterozygotes and elucidate the molecular relationship between SCA48 and SCAR16 diseases.

## 1. Introduction

Dominantly inherited spinocerebellar ataxias (SCAs) are a heterogeneous group of neurodegenerative disorders characterized by adult-onset progressive cerebellar ataxia. Many also have additional neurological symptoms including pyramidal and extrapyramidal features, and cognitive decline [1]. The global prevalence of SCA is estimated to be 1–5:10^5^, with the highest rates reported by population-based surveys in Portugal (5.6:10^5^), Japan (5:10^5^), and Southeast Norway (4.2:10^5^) [2,3].

Expanded polyglutamine repeats (CAG; polyQ) are responsible for the most common types of SCAs, followed by other noncoding repeat expansions and substitutions [1]. Recently, heterozygous pathogenic variants (mutations) in the *STUB1* gene (STIP1 homology and U–box containing protein 1; OMIM, 607207) causing a new form of SCA (SCA48) were described. The first reported Spanish family had a cognitive-affective syndrome (CCAS) and late-onset SCA following an autosomal dominant inheritance pattern [4]. Subsequent studies reported heterozygous *STUB1* mutations in several families with ataxia and cognitive–psychiatric disorder, often associated with other features such as dystonia, parkinsonism, chorea, and endocrine dysfunction [5,6,7]. To date, 18 heterozygous pathogenic *STUB1* variants have been reported to be associated with a SCA48 phenotype; however, how these mutations affect protein structure and the function remains unclear. Functional data are essential to elucidate mechanisms underlying the dominant inheritance of this disease [8,9].

The C-terminus of HSC70-interacting protein (CHIP) is a co-chaperone of E3 ubiquitin ligase encoded by the *STUB1* gene and acts as a component of the protein quality control system, targeting misfolded and damaged proteins for ubiquitination and subsequent degradation through proteasome and/or autophagy pathways [10]. The active CHIP functions as a dimeric protein consisting of an N-terminal tetratricopeptide repeat (TPR) domain (residues 26–127) mainly involved in the interaction of the CHIP with molecular chaperones, and a C-terminal U–box domain (residues 226–297) which functions as an E3 ubiquitin ligase. The two domains are separated by a central helical hairpin region (residues 128–225) that is critical for the stability and dimerization of the CHIP. The CHIP dimerization is also mediated by a second interacting surface in the U–box domain [11]. The CHIP is highly expressed in metabolically active tissues such as the brain and plays a protective role to prevent accumulation of abnormal proteins and neurodegeneration (10). Bi-allelic mutations in *STUB1* cause autosomal recessive spinocerebellar ataxia (SCAR16), an early-onset disorder with gait disturbance, dysarthria, head and hand tremor, hyperreflexia, cognitive decline, and, occasionally, hypogonadism [12]. SCAR16 variants are thought to destabilize the CHIP structure and lead to mutation-specific abnormalities such as decreased interaction with chaperones, reduced steady-state cellular levels, reduced ubiquitination activity, and/or misfolding [13,14].

In this study, we report three novel heterozygous *STUB1* variants in families presenting with SCA48 phenotypes. To understand the functional effects of these variants, we characterized the ubiquitin ligase activity and examined the structural properties of these variants using the recombinant CHIP purified from *Escherichia coli*. Our results suggest that the variants causing SCA48 are indistinguishable with regard to their functional impairment from previously described variants reported in families with SCAR16.

## 2. Results

### 2.1. Genetic Findings

Pedigrees of the three families with heterozygous *STUB1* variants segregating with dominant SCA48 are presented in Figure 1. Variants in *STUB1* (NM_005861.3) were detected by exome sequencing: in Family A, the proband, her father, and two of her three siblings carried the heterozygous variant c.152_158delinsCAGC p.(Arg51_Ile53delinsProAla) (abbreviated as R51_I53delinsPA). All carriers in Family A showed clinical characteristics similar to previously reported cases of SCA48 (Appendix A). The mother and one sister did not have symptoms of SCA48, and the sister was tested and did not carry the R51_I53delinsPA *STUB1* variant. The R51_I53delinsPA variant is not present in the Genome Aggregation Database (gnomAD) or in the dbSNP database, and the three amino acids located in the TPR domain (Arg51, Ala52 and Ile53) are well conserved throughout species. In addition to next-generation sequencing (NGS)-based gene panel analysis in all three affected siblings, the most severely affected family member, the father (II-2), was also tested for trinucleotide repeat expansions in other SCA associated genes (see Materials and Methods) with normal findings. The deceased paternal grandmother could not be tested, but according to the family, she also had ataxia and dysarthria.

The proband in Family B and her affected father carried the heterozygous *STUB1* variant c.426_441delinsT p.(Lys143_Trp147del) (abbreviated as K143_W147del). The variant is not present in gnomAD or in the dbSNP database. The five amino acids involved (Lys143, Lys144, Lys145, Arg146 and Trp147) are located in the central helical hairpin region and are well conserved throughout species. Both the proband and her father were also heterozygous for a 41-repeat expansion in the gene coding for TATA box binding protein, *TBP*.

The proband in Family C was heterozygous for a missense variant in *STUB1*: c.746G > T p.(Gly249Val) (abbreviated as G249V). Her mother died at age 73, not tested for the variant, and was reported to have behavioral variant frontotemporal dementia (bvFTD) from age 60. The variant is not present in gnomAD or in the dbSNP database, and Gly249, located in the U–box domain, is well conserved across species. No repeat expansions in other SCA-associated genes were found in this family.

### 2.2. Clinical Features

Seven patients were examined from three families, with the main clinical features summarized in Appendix A. The female proband in Family A (III-3) first developed dysarthria at the age of 40, and thereafter mild dysphagia. She was diagnosed with mild cerebellar gait ataxia at the age of 49. At age 51, she remains able to work part time. Her MRI showed cerebellar atrophy and four small high-signal lesions that have remained unchanged over time. Spinal fluid examination and evoked potentials were normal. The sister of the proband (III-2) developed depression at 47 years of age. Cognitive decline appeared some years later and, based on an MRI showing cerebellar atrophy, she was diagnosed with cerebellar cognitive affective syndrome (CCAS) at the age of 53. Other clinical features appeared normal in this patient. She is not able to work. The brother of the proband (III-1) presented with dysarthria at the age of 52. Eye movement examination showed dysmetric saccades. His gait and limb coordination were normal. His MRI showed clear cerebellar atrophy. The father of the proband (II-2) was diagnosed with cerebellar ataxia at 74 years of age, having presented with gait disturbance, encephalopathy, pyramidal signs, dysarthria, and dysphagia present from the age of 50. His MRI showed marked atrophy of the cerebellum and generalized cerebral atrophy. The unaffected sister (III-4) agreed to participate in the family study. Clinical examination produced normal findings, and her MRI was normal with no signs of cerebellar atrophy.

In Family B, the female proband (II-1) was first seen for dysarthria and unsteadiness at the age of 30. Examination revealed mild cerebellar dysarthria and mild tandem gait ataxia. She subsequently progressed with increasing gait ataxia which necessitated wheelchair use, encephalopathy, and bilateral horizontal nystagmus. An MRI taken at the age of 38 showed marked global cerebellar atrophy. The father (I-2) presented with gait disturbance initially thought to be parkinsonism at the age of 69 years and was later diagnosed with cerebellar gait disturbance at the age of 74. In addition to cerebellar atrophy, MRI findings revealed a small frontal lacunar infarct. The father is still ambulant and clinically less severely affected than his daughter.

The female proband of Family C (II-1) presented with cognitive and psychiatric symptoms associated with bvFTD disorder at the age of 48. She was found to have an ataxic gait, peripheral cerebellar dysfunction, and dysphagia, with progressive cerebellar atrophy on her MRI. The mother died at the age of 73 and was diagnosed with bvFTD at the age of 60. The mother’s sister died of dementia in her 50s, but the mother’s three siblings did not have any neurological symptoms. In Family C, only the proband was available for genetic testing.

### 2.3. Hsc70- and Self-Ubiquitination Activity of the CHIP Variants

The in vitro ability of the wild-type and variant CHIPs to ubiquitinate themselves with the Hsc70 substrate was investigated using recombinant maltose binding protein (MBP)–CHIP proteins and showed impaired ubiquitination activity toward the Hsc70 substrate for all the examined variants (Figure 2A). Self-ubiquitination activity appeared intact and similar to the wild-type control for the R51_I53delinsPA and K143_W147del variants as shown by the presence of several high-molecular bands, suggesting covalent attachment of ubiquitin to the CHIP itself; the G249V variant displayed impaired self-ubiquitination activity (Figure 2B).

### 2.4. Thermal Unfolding of the CHIP Variants

The effect of *STUB1* variants on the CHIP conformational stability as a function of temperature was examined by circular dichroism spectroscopy. The changes in protein molar ellipticity were monitored by gradually increasing the temperature from 20 °C to 90 °C (Figure 3). The thermal unfolding of the wild-type MBP–CHIP (Figure 3, blue) indicates a distinctive profile with three transitions representing denaturation of the MBP–CHIP dimers (phase 1), the MBP (phase 2), and the CHIP monomers (phase 3), as previously reported by us [13]. All three variants demonstrated a decrease in initial molar ellipticity at 222 nm, and the loss of one or two unfolding transitions. These results indicate that the mutations have a pronounced impact both on the structure of the MBP–CHIP and on the oligomeric states, likely due to altered tertiary structures. A reduction in signal was also detected for all the variants in their far-UV circular dichroism spectra (Figure A1), supporting the loss of secondary structures. These results suggest that all three mutations induce conformational changes in their encoded CHIP proteins, resulting in the acquisition of different oligomeric states and, thus, the abnormal thermal denaturation profiles (Figure 3).

### 2.5. Oligomerization States of the CHIP Variants

To elucidate the effect of *STUB1* variants on the native conformation and oligomerization states of the CHIP, we analyzed wild-type and mutant MBP–CHIP proteins by native polyacrylamide gel electrophoresis (native–PAGE) (Figure 4). Wild-type MBP–CHIP proteins were separated into three major bands, where the dominating band represented dimers, and the bands containing monomeric and higher order oligomeric structures appeared to a lesser extent. Examination of the mutant proteins showed a clear shift toward the higher-order oligomeric structures for all the variants, suggesting an increased propensity to form protein aggregates. While the amount of monomers remained similar for all the examined samples, dimeric structures were absent for the G249V variant, but detectable for R51_I53delinsPA and K143_W147del with 43% and 41% decreased intensity compared to the wild-type, respectively. Overall, these data are in accordance with the thermal unfolding profiles and indicate a profound alteration in the oligomeric structure of variants compared to the wild-type, shown by significant loss of dimers and higher-order oligomers comprising the major conformational state of mutant MBP–CHIP proteins.

## 3. Discussion

Heterozygous *STUB1* mutations cause dominantly inherited cerebellar ataxia SCA48, and despite its recent description, SCA48 has emerged as one of the more frequent subtypes of SCA [9]. Here, we report three novel heterozygous *STUB1* variants found in three families with SCA48 and describe the clinical variation associated with these variants. We also, for the first time, characterize the deleterious effect of heterozygous *STUB1* mutations on the CHIP protein structure and function.

Seven patients from three families were investigated. Clinically, all patients had a phenotype compatible with that described in SCA48 [4,5,16,17], and the majority of them had an early-onset dysarthria and progressive cerebellar ataxia with cognitive impairment developing later. Parkinsonism was seen in one patient. All patients developed their first symptoms at or after the age of 30. Despite a small sample size, we observed a tendency for females to develop a more severe phenotype, with earlier onset, than men. A sex-dependent penetrance has been suggested earlier [17], but also been questioned [18], and larger cohorts are probably needed to resolve this question.

Our functional data showed impaired ubiquitination activity on the Hsc70 substrate for all the CHIP variants, while the G249V variant was also unable to perform self-ubiquitination. The location of these mutations in functional CHIP domains suggests that the R51_I53delinsPA variant might interfere with the CHIP’s ability to target Hsc70 substrates at the TPR domain, while the G249V variant likely affects the CHIP ubiquitin ligase activity via the U–box domain, resulting in impaired ubiquitination of both the Hsc70 substrate and the CHIP itself. Further investigations into the conformational structure of the CHIP variants revealed significant alterations in protein oligomerization, demonstrated by the reduced amount of dimers and increased formation of higher-order oligomers or aggregates. Therefore, the mutations also appear to result in conformational changes that may contribute to pathogenesis in SCA48 patients through the gaining of toxic functions. This could, in particular, be the main mechanism for the K143_W147del and G249V variants, given the location of these mutations in the CHIP dimerization interfaces [19]. The dimerization of the K143_W147del variant is less severely affected, as a proportion of dimeric content is still seen. The complete lack of dimeric structures in the G249V variant, on the other hand, indicates an abolished dimerization interface in the U–box domain.

Similar defects in the protein structure and ubiquitination activity of the CHIP have also been described for bi-allelic *STUB1* variants associated with the SCAR16 disease [9]. In fact, our functional data on the SCA48 variants are highly similar to what we previously reported for the SCAR16 variant, p.Thr246Met [13]. Similar to the G249V variant, this mutation is also located in the U–box domain and displays impaired ubiquitination activity and a very small number of dimers, yet a higher degree of aggregates. In addition to the functional similarities, there is a clinical overlap between the recessive and dominant disorders, where ataxia and cognitive dysfunction are common in both groups. However, an earlier age of onset and additional multi-organ involvement defines the greater severity of SCAR16. This greater severity may reflect the lack of any functional allele in the recessive disease, as suggested in studies that show more severe physiological and behavioral defects in *Stub1 ^–/–^* animals rather than *Stub1^+/–^* animals [20,21].

The pathogenicity of heterozygous *STUB1* variants could reflect a dominant negative effect for SCA48 mutations. It remains unclear, however, why some *STUB1* mutations cause recessive ataxias with apparently unaffected heterozygous parents, while similar mutations cause dominant SCA48 in other families. The SCAR16 variant p.Asn65Ser (N65S) was originally described as a cause of SCAR16 in three siblings born from healthy parents [22]. This variant was, however, was recently reported as a cause for an autosomal dominant disorder in three families, with later onset and different clinical picture compared to the original SCAR16 families [17]. Based on this, we suggest that a close follow up of *STUB1* carriers in SCAR16 families is warranted. Penetrance may possibly vary to a greater degree in dominant forms, and it is probable that genetic variation elsewhere in the genome impacts the penetrance or expression of certain variants. For example, an additional variant of unknown significance in *PRKCG*, the gene coding for protein kinase C Gamma associated with SCA14, was seen in heterozygous carriers of N65S, suggesting a possible synergy of the two variants in the early development of disease [17]. In our Family B, both the proband and her father were heterozygous for a 41-repeat expansion in *TBP*. Repeat expansions of 41–45 in *TBP* have been described as intermediate penetrant alleles associated with SCA17 [23,24]; however, the pathogenicity of the 41-repeat expansion allele has been questioned since it is present at a minor allele frequency of 0.5–0.7% in control populations [25,26] and has been found in many asymptomatic individuals [27]. Since no SCA17 families with a heterozygous 41-repeat allele and dominant inheritance have been published, it is highly unlikely that this alone can cause the severe phenotype in patient II-1, and milder effects on I-2 in family B. We cannot, however, exclude a modifying effect.

In summary, our findings demonstrate similar functional and biochemical properties for the heterozygous *STUB1* variants identified in this study and those of previously reported recessive variants associated with SCAR16. The reduced penetrance observed in some of the reported SCA48 families might suggest the presence of additional undiscovered genetic factors in these families. Our data also suggest that *STUB1* carriers in SCAR16 families require regular clinical follow-up. Future studies are required to elucidate the exact genotype–phenotype correlation as well as pathways behind the pathogenesis of these variants.

## 4. Materials and Methods

### 4.1. Clinical Data

Three families with seven patients were included in this study, after having the results from diagnostic NGS gene panel analysis requested by a neurologist (Family A and B) or geneticist (Family C). The three siblings in Family A were referred by three different neurologists as separate probands, at that time not aware of the affected siblings. Clinical information regarding the patients and MRI scans is summarized in Appendix A. All patients were investigated by at least one neurologist. Blood samples were collected from all available family members and DNA purified by standard methods. Written informed consent was obtained from all participants. All patients and involved families were offered genetic counselling.

### 4.2. Genetic Analyses

In Families A and B, a targeted 639 gene panel, including ataxia genes, was performed and extracted from whole exome sequencing data in III-1, III-2, and III-3 in Family A and II-1 in Family B. Whole exome sequencing was performed using genomic DNA from the probands. DNA samples were prepared using the SeqCap EZ MedExome target enrichment kit (Roche NimbleGen, Madison, WI, USA) and then underwent paired-end 150 nucleotide sequencing on the Illumina NextSeq500 (Illumina, Way San Diego, CA, USA). Alignment and variant calling were performed as previously described [28,29]. Data annotation and interpretation were performed using the Cartagenia Bench Lab, NGS module (Cartagenia, Leuven, Belgium). The *STUB1* variants in Families A and B were verified by Sanger sequencing in the probands and in affected family members. The affected fathers (I-2) in both Family A and B were not analyzed by NGS, only by Sanger sequencing of the relevant *STUB1* variant. In patient II-1 from Family C, exome sequencing was performed by the Australian Genome Research Facility Ltd., with the SureSelect XT Low Input Clinical Research Exome V2 (Agilent Technologies, CA, USA) kit and sequencing on an Illumina NovaSeq6000. Data were processed using the GATK best-practice pipeline v4.0 (Broad Institute, Cambridge, MA, USA), including alignment to the GRCh38 reference genome and variant calling using HaplotypeCaller. Variant prioritization using a 116 gene target panel (including adult-onset ataxia genes and early-onset dementia genes) was performed using Alissa Interpret v5.2.7 (Agilent Technologies, CA, USA). Patient II-2 in Family A, patients I-2 and II-1 in Family B, and patient II-1 in Family C were tested for trinucleotide repeat expansions in *ATXN1* (SCA1), *ATXN2* (SCA2), *ATXN3* (SCA3), *CACNA1A* (SCA6), and *ATXN7* (SCA7). Further, patient II-2 in Family A and patients I-2 and II-1 in Family B were also tested for repeat expansions in *PPP2R2B* (SCA12), *TBP* (SCA17), *ATN1* (DRPLA), and *FXN* (FRDA). Patient II-1 in Family C was also tested for expansions in *HTT*. *FMR1* repeat expansions were excluded in all four affected patients in Family A, and in the father I-2 in Family B. In patient II-1 in Family B, whole genome sequencing (WGS) was performed at the Genomics Core Facility (University of Bergen, Bergen, Norway) (only looking at the *STUB1* gene) to search for a second *STUB1* variant (deletion or deep intronic variant), with no second finding. WGS was performed as previously described [28]. Furthermore, haplotype analyses revealed that the three affected siblings in Family A did not share the same maternal *STUB1* allele.

### 4.3. In Vitro Expression of Mutant CHIP Proteins

The full-length wild-type *STUB1* cDNA was previously cloned into the bacterial expression vector His6-MBP-pETM-41 [13]. The plasmid was used to generate *STUB1* constructs containing c.152_158delinsCAGC, c.426_441delinsT and c.746G > T by QuikChange XL Site-directed Mutagenesis Kit (Agilent Technologies) (mutagenesis primers available upon request). Wild-type and mutant MBP–CHIP fusion proteins were further expressed overnight at 25 °C in BL21-CodonPlus (DE3)-RP cells (Agilent Technologies). The His-tagged proteins were then purified by Ni-NTA agarose nickel affinity binding resin (QIAGEN, Hilden, Germany), following the manufacturer’s instructions. The CHIP proteins were stored in 100 mM HEPES, 100 mM NaCl, 5 mM DTT, and 10% glycerol at −80 °C.

### 4.4. In Vitro Ubiquitination Activity Assay

An in vitro ubiquitination activity assay was set up for the wild-type and mutant MBP–CHIP proteins in a 30 μL reaction containing 2.5 μg CHIP and 1 μg His-HSPA8 /HSC70 (LifeSpan BioSciences, Seattle, WA, USA) incubated with 250 μM ubiquitin (Boston Biochem, MA, USA), 2.5 μM UbcH5c (E2) (Boston Biochem), 0.05 μM Ube1 (E1) (Boston Biochem), 50 mM Tris HCl, pH 7.5, 0.6 mM DTT, and 2.5 mM Mg-ATP for 1 h at 37 °C. The negative control samples were prepared using wild-type CHIP proteins and without adding ubiquitin to the reaction mixture. Samples were analyzed for CHIP and Hsc70 ubiquitination by sodium dodecyl sulfate (SDS)-PAGE and Western blotting using anti-HSC70/HSP73 (1:10,000, Enzo Life Sciences ADI-SPA-815, UK) and anti-CHIP (1:2000, Sigma C9243, MO, USA) antibodies.

### 4.5. Circular Dichroism Spectroscopy

Circular dichroism far-UV and thermal denaturation measurements were recorded using a Jasco J-810 spectropolarimeter equipped with a CDF-426S Peltier temperature control unit (Jasco Products, OK, USA). The wild-type and mutant MBP–CHIP proteins were prepared at 2.5–5.8 µM concentration in a buffer containing 10 mM potassium phosphate and 100 mM sodium fluoride at pH 7.4. The protein concentrations were determined using a NanoDropTM One Microvolume UV–Vis spectrophotometer (Thermo Scientific, MA, USA), with theoretical extinction coefficients ε_280_ = 96,720 (wild-type MBP–CHIP, G249V, and R51_I53delinsPA) and 91,220 (K143_W147del) M^−1^cm^−1^. Far-UV spectra were acquired in the range of 185–260 nm at a scan rate of 50 nm/min at 20 °C, using a quartz cell with a path length of 1 mm. Three scans were accumulated for each spectrum and three spectra were buffer subtracted and then averaged. Thermal denaturation profiles were obtained by recording the decrease in ellipticity at 222 nm as a function of temperature in the range of 20–90 °C with a scan rate of 40 °C/h. The results are expressed in mean residue ellipticity [θ]_mrw_ = θ(deg × cm^2^ × dmol^−1^), and BeSTSel (Beta Structure Selection) was used to estimate the secondary structure content [30]. Final graphs were prepared by using GraphPad Prism software (San Diego, CA, USA).

### 4.6. Native Polyacrylamide Gel Electrophoresis (Native–PAGE)

In total, 5 µg of wild-type and mutant MBP–CHIP in Native Sample Buffer (Bio-Rad Laboratories, CA, USA) was loaded in a 10% Mini-PROTEAN^®^ TGX^TM^ Precast Protein Gel (Bio-Rad Laboratories) and run at 4 °C for 3 h at 140 V in a Tris/glycine buffer (25 mM Tris, 192 mM glycine) at pH 8.3 as running buffer. The gel was stained using Coomassie Brilliant Blue G-250.

## Figures and Tables

**Figure 1 ijms-22-05870-f001:**
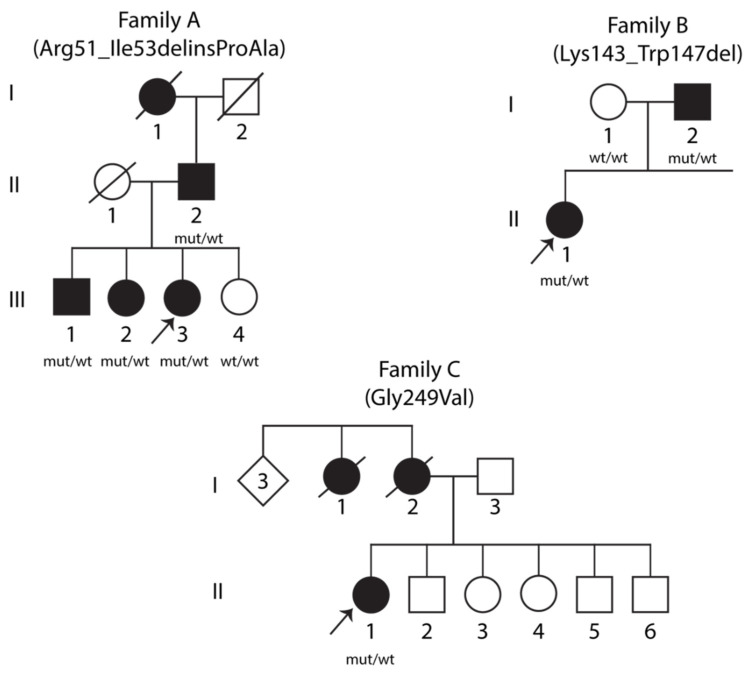
Pedigrees of three families with SCA48. Clinically affected individuals are indicated by filled symbols. Arrows denote the probands. Genetic statuses of examined individuals are provided below the symbols, where “mut” represents the mutant allele and “wt” represents the wild-type allele.

**Figure 2 ijms-22-05870-f002:**
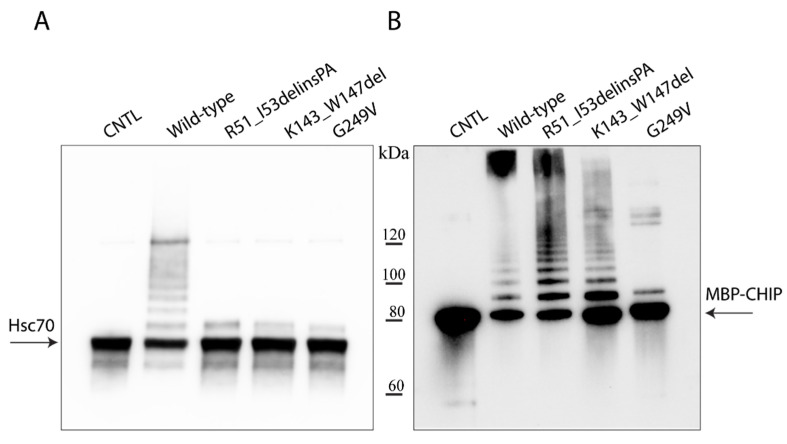
Ubiquitination activity of wild-type and mutant MBP–CHIP proteins. In vitro ubiquitination activity test was performed on MBP–CHIP recombinant proteins using Hsc70 (**A**) and MBP–CHIP (**B**) as the substrates. A separate wild-type reaction without ubiquitin was used as the negative control (CNTL) for each assay.

**Figure 3 ijms-22-05870-f003:**
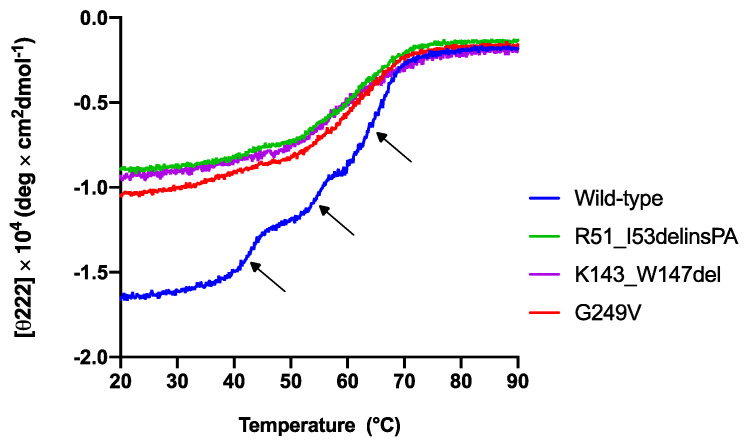
Thermal unfolding profiles of wild-type and mutant MBP–CHIP proteins monitored by circular dichroism spectroscopy. Thermal unfolding curves were obtained for MBP–CHIP proteins by following changes in molar ellipticity at 222 nm wavelength as a function of temperature. Wild-type MBP–CHIP shows three distinct transitions at approximately 42.8 °C, 54.6 °C, and 65.1 °C, as indicated by arrows, whereas all three variants show loss in both ellipticity and transitions.

**Figure 4 ijms-22-05870-f004:**
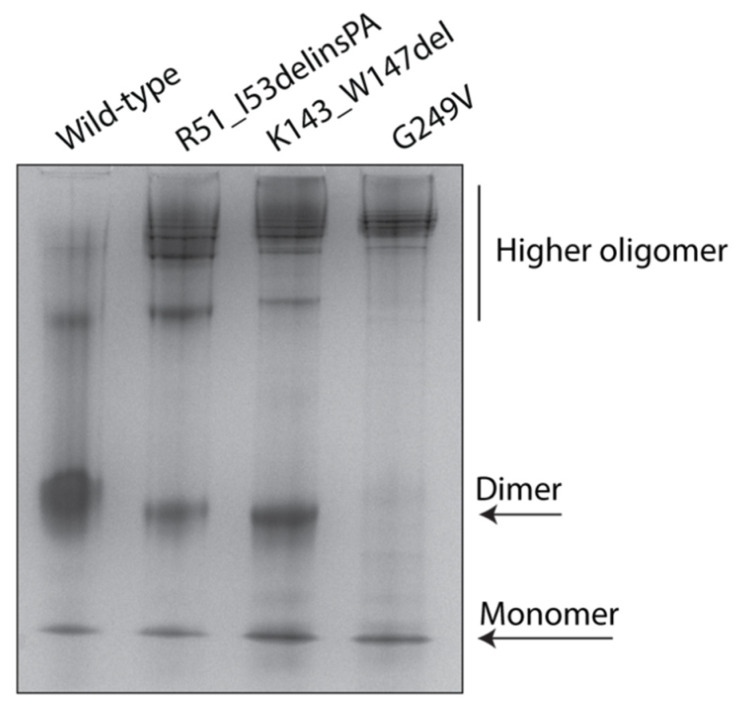
Oligomerization states of wild-type and mutant MBP–CHIP proteins. Native structures of 5 µg MBP–CHIP proteins were studied by native–PAGE on a 10% gel. Protein bands were visualized by Coomassie blue staining and analyzed by Fiji (ImageJ) software [15].

## Data Availability

Data underlying this article can be made available upon request.

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
