# Peer review of "Genetic Dominant Variants in STUB1, Segregating in Families with SCA48, Display In Vitro Functional Impairments Indistinctive from Recessive Variants Associated with SCAR16"

_ijms, 2021, doi:10.3390/ijms22115870_

Round 1
Reviewer 1 Report
This was a genetic study on three autosomal dominant cerebellar ataxia families having three different heterozygous mutations in STUB1 gene. The authors identified three new variants and assessed their functional consequences in vitro using recombinant C-terminus of HSC70-interacting protein and CD spectra. They found that these mutations all show impaired ubiquitin ligase activity and altered oligomerization properties, which led them to conclude that these mutations are responsible for SCA48, a dominantly-inherited ataxia with cerebellar cognitive affective syndrome.
Authors also suggest further investigations are needed to verify the disease status of SCAR16 heterozygotes and elucidate molecular relationship between SCA48 and SCAR16 diseases.
I feel that this is a sound study with appropriate discussion.
I have only one suggestion. In Figure 3, it would be more comprehensive if the authors could points three arrows that indicate transitions.
Author Response
We are grateful for the positive comments and thank the reviewer for the suggestion. We have now included three arrows indicating the three temperature transitions in Figure 3 as requested.
Reviewer 2 Report
Authors describe three novel heterozygous STUB1 variants identified in three families affected with late onset ataxia. They also investigate the effect of these variants on STUB1 structure and function. The study is of interest even though few points should be clarified/discussed
Major revision
-I suggest adding an assessment of the degree of cerebellar atrophy.
-Can the Authors compare the degree of cerebellar atrophy of SCA48 patients with that of SCA16 patients?
-A wide intrafamilial and intergenerational variability has been reported and in particular individuals from the second generation seem to have a more complex phenotype, with a lower age at onset. The Authors should discuss this point.
Author Response
We are grateful to this reviewer for their interest and for the clinical points raised. With regard the degree of cerebellar atrophy, we did not performed volumetric analysis. We do, however, provide a crude assessment in Table 1 that identifies those with most marked cerebellar loss. In addition, we have discussed the MRI findings in the text.
We assume that this reviewer means SCAR16 and not SCA16 (which is now SCA15). Comparison of MRI images taken at various times and varying stages of disease is not straightforward. The SCAR16 patients we described previously had changes compatible with hypoplastic cerebellum, while others SCAR16 patients described in the literature have varying degrees of cerebellar atrophy that are not quantified volumetrically. Based on our own material, we cannot make any meaningful comparison.
Regarding intergenerational variability in SCA48 families and the possibility that individuals from the second generation have a more complex phenotype, this is an interesting hypothesis. To our knowledge, the literature does not suggest anticipation in STUB1 related disease, with earlier and more severe disease in subsequent generations. This effect usually attributed to dynamic mutations such as trinucleotide expansions, which is not the case here. In our dataset, we believe that we have too few family members to draw such a conclusion. In family A, the father of the proband (II-2) was diagnosed with cerebellar ataxia at 74 years of age having presented with gait disturbance, encephalopathy, pyramidal signs, dysarthria, and dysphagia present from the age of 50. His first symptoms were present from the age 50, i.e. in the same age range as other affected members in the family (range 40-52 yrs). Initially, the father in family A was thought to have an alcohol related problem, and while this was not confirmed, it did delay the molecular diagnosis in him (age 74). We have insufficient information on his affected mother to make a comparison. In family B, the father is less affected than his daughter. However, since we have not been able to investigate or test the siblings of the proband, we can not at this point conclude that we see a second generation effect in this family. In family C, the proband was the only family member that was genetically tested. A sex dependent penetrance seen in SCA48 families have been suggested by others (Reference: PMID: 32713943 (ref 17 in manuscript)). Our cohort is too small to suggest such an association, as we also have discussed in the manuscript. It will be interesting in the future to investigate the exact causes of the different penetrance observed for SCA48 families.
Reviewer 3 Report
This article described a good mechanism-based genotype to phenotype relationship on clinical discovered genetic diseases. Currently only few disease-causing variations could be proved by wet-lab in vitro biochemistry assays.
The authors should emphasize 3 novel variations were not described in current build of dbSNP.
The authors are suggested to demonstrate whether mixture of WT and mutant purified protein can abolished the normal E3 activity as the evidence of autosomal dominant phenotype of SCA48.
Author Response
We thank this reviewer for their positive response to our work. We have now included text in the manuscript (Section 2.1 Genetic findings, page 5) stating that none of the variants was found in the dbSNP database.
We appreciate this relevant question from the reviewer regarding whether mixture of WT and mutant purified protein can abolished the normal E3 activity. We have previously examined the ubiquitin ligase activity of recombinant MBP-CHIP proteins as mixtures of wild-type and mutant proteins at the ratios of 3:1, 3:2, 3:3 and 3:4 (data not shown). Unfortunately, our data did not show consistent results. In the future, we therefore plan to examine the dominant negative effect of SCA48 mutations by using other expression systems such as cells co-transfected with the wild-type and mutant CHIP plasmids. However, these experiments are time consuming and it is not possible to carry out due to the strict timeline for the resubmission of this manuscript.